# Determinants of utilization of cervical cancer screening among women in the age group of 30–49 years in Ambo Town, Central Ethiopia: A case-control study

Dereje Lemma, Mecha Aboma [iD]*, Teka Girma, Abebe Dechesa

Ambo University College of Medicine and Health Sciences, Ambo, Ethiopia

* abomamecha@gmail.com, mecha.aboma@ambou.edu.et

## Abstract

### Background

Globally, cervical cancer is the second most common and the leading cause of death in women in low-income countries. It is one of the potentially preventable cancers, and an effective screening program can result in a significant reduction in the morbidity and mortality associated with this cancer; however, evidence showed that only a small percentage of the women were screened. As a result, predictors of cervical cancer screening usage among women in Ambo town, central Ethiopia, were identified in this study.

### Method

Unmatched, a community-based case-control study was conducted among 195 randomly sampled women in the age group of 30–49 years in Ambo town from February 1 to March 30, 2020. Data was collected using an interviewer-administered questionnaire. Descriptive, bivariate, and multivariable binary logistic regression analysis was done using SPSS.

### Results

A total of 195 study participants, sixty-five cases and one hundred thirty controls, participated in this study, making a response rate of 100%. Being in the age group of 30–34 years old (AOR = 0.2; 95% CI: 0.06–0.7), being Para five and above (AOR = 4.5; 95% CI: 1.4–14.1), modern contraceptive utilization (AOR = 5.4; 95% CI: 1.8–16.3) and having high-level knowledge regarding cervical cancer screening and its predisposing factors (AOR = 5.9; 95% CI: 2–17) were significantly associated with the utilization of cervical cancer screening.

### Conclusion

The age of women, parity, use of modern contraception, and level of knowledge regarding cervical cancer screening and its predisposing factors were the determinants of the utilization of cervical cancer screening among women. As a result, the media, the health bureau,

**Data Availability Statement:** The dataset/ SPSS data file used and analysed throughout the present study is uploaded based on your request.

**Funding:** The authors received no specific funding for this work

**Competing interests:** The authors declare that no competing interests

and health professionals should advocate raising awareness about cervical cancer and its preventative methods, which are primarily focused on screening.

## Introduction

Cervical cancer is a slow-onset malignancy found in the interior lining of the cervix, at the junction of the vagina and uterus [1]. Cervical cancer continues to be a major public health problem affecting middle-aged women globally, with around 570,000 cases of cervical cancer and 311,000 deaths occurring in 2018 [2]. Similarly, according to global cancer statistics, cervical cancer ranks fourth for both incidence (6.6%) and mortality (3.5%) among females in 2018 [3]. Every two minutes, a woman dies from cervical cancer, with more than 90% of these deaths accounting for low- and middle-income (LMIC) countries [4].

In less developed regions, cervical cancer makes up 85% of the new cases and 87% of the deaths that occur with the second most commonly diagnosed cancer and the third leading cause of cancer death among females [5, 6]. Similarly, in Sub-Saharan Africa, cervical cancer incidence has been increasing, becoming the second most prevalent and incidental type of cancer among women after breast cancer [7].

In Ethiopia, cervical cancer is the most common (31.8%) diagnosed cancer among all cancer cases and the second leading gynecologic cause of death, next to breast cancer, among women aged 15–44 years in 2012 [8]. Every year in Ethiopia, 7095 women are diagnosed with cervical cancer and 4732 die from the disease, putting a 29.4 million population at risk for cervical cancer in 2012 [9]. By the year 2030, cervical cancer will kill more than 443,000 women yearly worldwide, most of them in sub-Saharan Africa, including Ethiopia [10].

Cervical cancer is robbing us of our mothers, daughters, sisters, and grandmothers, impacting our communities and threatening the social and economic fabric of society. Even though cervical cancer is preventable, treatable, and the only cancer with a clear path to elimination, thousands of women are dying unnecessarily in the prime of their lives from this preventable and treatable disease [4]. In Sub-Saharan Africa, approximately 80–90% of women have never had a pelvic examination, and less than 5% have access to screening [11].

Regular screening is associated with a 67% reduction in stage 1A cancer and a 95% reduction in stage 3 or worse cervical cancer and prevents 70% of cervical cancer deaths (at all ages). Also, if everyone attended screening regularly as recommended, 83% of cervical cancer could be prevented [12].

Cervical cancer mostly develops slowly; it usually takes ten years to become cervical cancer, and when detected early as a precancerous lesion, it can be treated effectively [13]. However, due to a lack of an effective prevention mechanism, the majority of cervical cancers (more than 80%) in Sub-Saharan Africa are detected in late stages, which is associated with low survival rates after surgery or radiotherapy [14]. Cytology-based screening (Pap smear test) has significantly reduced cervical cancer incidence and mortality in developed countries. Yet it has had limited success in Ethiopia and other resource-poor countries, as it requires repeated testing, laboratory analysis, and proper diagnostic, treatment, and follow-up protocols [15, 16]. Visual Inspection with Acetic Acid (VIA) is an evidence-based and affordable alternative approach for cervical cancer screening in low-resource settings. Studies have reported VIA sensitivity for detecting precancerous lesions comparable to or greater than cervical cytology, while requiring fewer resources and being feasible to carry out in low-level health facilities [15].

Visual Inspection with Acetic Acid combined with cryotherapy, ideally in a single visit approach (SVA), is an effective and efficient strategy for the prevention of cervical cancer in

low-resource settings, and can be conducted by competent clinicians and nurses [17]. Visual Inspection with an Acetic Acid-based program using the SVA strategy has been shown to significantly reduce precancerous lesions of the cervix, cervical cancer incidence, and mortality [18].

The human papillomavirus (HPV) is a significant co-factor to cervical cancer. The development of vaccines against HPV has been a major advance for the prevention of this cancer. Nevertheless, large-scale implementation of HPV vaccination is still lacking in developing countries and will not replace the need for cervical cancer screening [19]. Cervical cancer prevalence is one of the disease spectrums that determine the country's socioeconomic status, as well as the difference between the upper and lower classes within the country. The majority of these deaths can be prevented through universal access to comprehensive cervical cancer prevention and control programs, which have the potential to reach all girls with human papillomavirus (HPV) vaccination and all women who are at risk with screening and treatment for pre-cancer [16].

In Ethiopia, routine access to cervical cancer screening was not available and treatment of precancerous cervical lesions didn't exist until the implementation of the Addis Tesfa project in 2009 [20]. According to the Information Centre on HPV and Cancer 2017, in Ethiopia, the overall coverage of cervical cancer screening was found to be 0.8%, with only 0.6% of all women, 1.6% of urban women, and 0.4% of rural women aged 18–69 years screened every three years [15, 21]. The government of Ethiopia launched screening for cervical cancer in 2014, and the VIA recommended for those women between the ages of 30 and 49 within every five years. Despite the fact that there are guidelines, protocols, and instruments available for those with a precancerous lesion, only a small number of women are screened for cervical cancer [21].

To eliminate cervical cancer, WHO targeted three goals: goal one says 90% of girls have to be fully vaccinated with the HPV vaccine by age 15, goal two 70% of women should be screened with a high-performance test by the age of 35 and again at 45, and goal three, 90% of women identified with cervical cancer should receive treatment (90% of women with pre-cancer treated; 90% of women with invasive cancer managed) that is designed to eliminate cervical cancer by reducing the number of cases to 4 cases per 100,000 women per year [4]. Similarly, the Ethiopian Federal Minister of Health aimed for at least 80% coverage of the target populations for pre-invasive cervical cancer screening and treatment by 2020 [22].

But a community-based cross-sectional survey of nine regions, including Addis Ababa city administration and Dire Dawa administration, shows an extremely low rate of cervical cancer screening (2.9%) [23]. Likewise, a study conducted in different parts of the country shows very low utilization of cervical cancer screening [24, 25]. Even a study conducted among female health care providers showed that only 17% of them had ever been screened for cervical cancer [26].

Poor access to screening and treatment services is attributed to more than 85% of women's deaths in low and middle-income countries [27]. Most cervical cancer screening studies done in the country don't elucidate factors affecting the utilization of cervical cancer screening. As a result, little is known about the determinants of cervical cancer screening utilization among women aged 30–49 years in Ambo and throughout the country [8, 28]. As a result, predictors of cervical cancer screening usage among women aged 30–49 years in Ambo Town were identified in this study.

## Methods

### Study setting

This study was conducted in Ambo town, West Shoa Zone, Oromia Regional State, central Ethiopia from February 1–March 30, 2020. Ambo town is found at a distance of 114 km from the capital city of Ethiopia, Addis Ababa, in the west direction. The town has three urban and

three rural administrative kebeles. A kebele is the smallest administrative unit in Ethiopia. According to Ambo Town Administration Office 2018 data, the total population of the town was 108,000, of which 53,400 males and 54,600 females. The town has one referral hospital, one general hospital, two health centers, nine health posts, and twenty one private clinics. Ambo General Hospital was the only hospital providing cervical cancer screening during this study was conducted. Ambo General Hospital had 287 health care workers. Those were 8 special doctors, 19 general practitioners, 69 nurses, and other health care providers [29].

## Study design, sample size, and sampling procedures

A community-based unmatched case-control study was conducted to identify factors that determine the utilization of cervical cancer screening among women in the age group of 30–49 years. All women in the age group of 30–49 living in Ambo town selected by simple random sampling techniques and available during the data collection periods were the sampling unit included in the study. The study excluded psychiatric and critically ill patients, as well as women who had already been diagnosed with cervical cancer and were receiving treatment. All randomly selected women in the age group of 30–49 years and residents of the study area who were screened for cervical cancer as cases and women of the same age group who didn't get screening for cancer as controls were the study unit of this study.

The sample size for this study was determined by using a double population proportion formula using EPI-info version 7.2 with the assumption of power = 80%; confidence level = 95%, case to control ratio of 1:2, P1 = the proportion of women in the age group of 30–49 years with parity five and above screened for cervical cancer, and P2 = the proportion of women in the age group of 30–49 years with parity five and above not screened for cervical cancer, as the main predictors of the outcome, which was 2.1% and 13.0% among cases and controls, respectively. Finally, by considering 5% non-response rates, a total of 195 (65 from cases and 130 from controls) were generated, the largest sample size [24].

Cases were selected in Ambo General Hospital from the cervical cancer screening registration book by using a simple random sampling technique, and a list of selected study participants, their address, and phone number were taken from the registration book. Controls were selected from the nearest neighbors to the cases, and then the same interview questionnaire was administered to both cases and controls at their household.

## Data collection tool and personnel

Data were collected by four trained BSc nurses under the supervision of two health officers using a pretested structured interviewer-administered questionnaire adapted and modified from similar literature [30–33, 39]. The questionnaire asks about sociodemographic factors, reproductive-related factors, knowledge questions with correct and incorrect answers, and attitudes about cervical cancer screening on a five-point Likert scale (1-strongly agree, 2-agree, 3-neutral, 4-disagree, and 5-strongly disagree).

## Data management and analysis

Data quality was assured through pre-testing the data collection tools on 5% of the total sample size before it was used for the actual data collection in a similar population who were not included in the study subjects. Data collectors and supervisors were trained for one day by the principal investigator on the study instruments and consent form, how to interview, and data collection procedures. The data collection processes were closely supervised by supervisors and investigators. Before data entry, the questionnaires were checked for completeness, consistency, and correction measures made by supervisors and investigators.

The data were then coded, entered into EPI-Data 3.1, and exported to SPSS software version 25 for data processing, cleaning, and analysis. Descriptive analysis like frequency and percentage was carried out to describe sociodemographic characteristics, reproductive, knowledge, and attitude-related determinants of utilization of cervical cancer screening among women, and results were presented in texts, tables, and graphs. The bivariate and multivariate analyses were done using binary logistic regression to identify factors associated with the utilization of cervical cancer screening among women. Candidate variables for the final model (multivariate binary logistic regression) were identified using a binary logistic regression model at a p-value of less than 0.25, and the final model multiple logistic regression was done to see the independent effect of each explanatory variable on the study variable at a p-value of less than 0.05.

The Hosmer and Lemeshow goodness-of-fit (P-value = 0.348) was checked to test for model fitness. The independent variables were tested for multi co-linearity using the Variance Inflation Factor (VIF) and the Tolerance tests, and no variables were found to have a VIF greater than 2 to be omitted from the analysis.

## Terms and operational definition

**Cases:**—Women who were screened for cervical cancer in the age group of 30–49 years.

**Controls:**-Women who didn't screen for cervical cancer in the age group of 30–49 years.

**Knowledge:**—It is information that an individual has become aware of what cervical cancer screening is and factors that predispose to it. In this study, it was measured based on the ability of the respondents to correctly answer symptoms, risk factors, and preventive measures for cervical cancer and cervical cancer screening.

It was assessed using six items, each having correct and incorrect responses. Each item contains 1 point for a correct response, 0 for an incorrect response, and I don't know. The maximum correct response contains 6 points and a minimum of zero. The knowledge of the study participants toward cervical cancer screening was assessed using the sum score of each item based on Bloom's cut-off point [34]. The scores were classified into 3 levels as follows:

**High-level of knowledge:**-Knowledge scores that fell between 4. 8–6 (80%-100%).

**Moderate level of knowledge:**-Knowledge scores that fell between 3. 6–4. 7 (60%-79.9%).

**Low-level of knowledge:** - Knowledge score of less than 3. 5 points ($< 60\%$).

**Attitude:**—Includes 8 items to assess the perception or outlook regarding causative factors and preventive measures of cervical cancer screening. All individual answers were summed up for total scores and calculate for means percent. The scores were classified into 3 levels (Positive Attitude, Neutral Attitude and Negative Attitude) according to Bloom's cut-off point.

**Positive attitude:**—Attitude scores that fell between 6. 4–8 (80% - 100%).

**Neutral attitude:**—Attitude scores that fell between 4. 2–6. 3 (60% -79. 9).

**Negative attitude:**—Attitude score less than $<4.2$ ($<60$).

**Modern contraceptives:**—those women who used any type of modern contraceptives up to now which include Depo-Provera, pills, copper IUCD, ligation and implants.

## Patient and public involvement

There was no patient or public involvement in this study. Patients were not requested to comment on the study design and were not involved in developing patient-relevant outcomes or interpreting the results. Patients were not involved in the development of the dissemination strategy.

## Results

### Socio-demographic characteristics of respondents

A total of 195 study participants, sixty-five cases and one hundred thirty controls, participated in this study, making a response rate of 100%. The majority of respondents were between the ages of 35–39 years (44.6%) for cases, while between 30–34 years of age (61.5%) for controls, and the median age for study participants (cases and controls) was 34 years. The mean age for

**Table 1. Socio-demographic characteristics of women in the age group of 30–49 years in Ambo town, Oromia Regional State, Ethiopia, February to March 2020.**

| Socio-demographic variables of study participants (n = 195) | Frequency | |
| --- | --- | --- |
| | Number/percentage of cases (n = 65) | Number/percentage of controls (n = 130) |
| **Age (in years)** | | |
| 30–34 | 19 (29.2) | 80 (61.5) |
| 35–39 | 29 (44.6) | 33 (25.4) |
| ≥40 | 17 (26.2) | 17 (13.1) |
| **Religion** | | |
| Orthodox | 25 (38.5) | 38 (29.2) |
| Protestant | 34 (52.3) | 84 (64.6) |
| Muslim | 4 (6.2) | 3 (2.3) |
| Wakefatu | 2 (3.1) | 3 (2.3) |
| Others * | 0 (0.0) | 2 (1.5) |
| **Ethnicity** | | |
| Oromo | 64 (98.5) | 124 (95.4) |
| Amhara | 1 (1.5) | 6 (4.5) |
| **Marital status** | | |
| Married | 56 (86.2) | 110 (84.6) |
| Others ** | 9 (13.8) | 20 (15.4) |
| **Educational status** | | |
| Cannot write or read | 14 (21.5) | 17 (13.1) |
| Primary (1–8) | 14 (21.5) | 27 (20.8) |
| Secondary (9–12) | 13 (20) | 27 (20.8) |
| Diploma and above | 24 (36.9) | 59 (45.4) |
| **Occupation** | | |
| Unemployed | 37 (56.9) | 56(43.1) |
| Government employee | 21 (32.3) | 49 (37.7) |
| Self-employee | 7(10.8) | 25 (19.2) |
| **Monthly income** | | |
| <1500 ETB | 22 (33.8) | 30 (23.1) |
| 1500–3000 ETB | 7 (10.8) | 20 (15.4) |
| >3000 ETB | 36 (55.4) | 80 (61.5) |

Others * Catholic (1), Adventist (1) others

** widow (8), single (13), separated (5), (ETB) Ethiopian Birrs.

cases and controls was 37.1 (±4.76 SD) and 34.3 (±4.76SD) respectively. The majority of study participants, 34 (52.3%) of cases and 84 (64.6%) of controls, were protestant religious followers. A larger proportion of the cases, 64 (98.5%), and controls, 124 (95.4%), were Oromo in their ethnicity. About 24 (36.9%) and 59 (45.4%) of cases and controls had diplomas and above educational levels, respectively (Table 1).

## Factors related to the reproductive health of study participants

The majority of both groups of study participants saw their first menstrual cycle at an age of fewer than 15 years, 51 (78.5%) and 98 (75.4%), and the mean age at their menarche (first menses) was 13.49 (±1.05) and 13.60 (±1.27) years for cases and controls, respectively. In the majority of the studies, participants started their first sexual intercourse at the age of 18 and above, 51 (78.5%) for cases and 102 (84.3%) for controls.

In the majority of both groups of study participants, their parity was less than five, 40 (65.6%) and 97 (89.8%) for cases and controls, respectively, and about 38 (62.3%) of cases and 69 (63.2%) of controls gave their first delivery in the age group of 20–24 years. About 51 (78.5%) of cases and 75 (57.7%) of controls used modern contraception, while the majority of respondents in both groups, 62 (95.4%) of cases and 124 (96.1%) of controls, never used condoms.

The majority of respondents' husbands, 53 (94.6%) among cases and 102 (92.7%) among controls, had no other wife, while 58 (89.2%) of cases and 110 (90.9%) of controls had only one partner in their time life (Table 2).

## Knowledge related factors of respondents

Thirty-six (55.4%) of the respondents have high-level comprehensive knowledge among cases, whereas 23 (17.7%) of the respondents among control have high-level knowledge-based on blooms cut-off point (Fig 1).

## Factors related to the attitude of study participants

Thirty-six (55.4%) of the respondents have a positive attitude toward cases; ninety-one (70%) of the respondents have a positive attitude toward controls based on bloom cut-off point (Fig 2).

## Determinants of utilization of cervical cancer screening

The result of backward likelihood multivariate logistic regression analysis revealed that only age, parity, use of modern contraception, and level of knowledge showed statistically significant associations with the utilization of cervical cancer screening among women, after controlling for potential confounders.

Hence, the odds of utilization of cervical cancer screening among women in the age group of 30–34 years were 0.2 times less as compared to women in the age group of 40 years and above (AOR = 0.2; 95% CI: 0.06–0.7)

The odds of using cervical cancer screening among Para five and above women were 4.5 times higher when compared to women who were less than Para five (AOR = 4.5; 95% CI: 1.4–14.1). Women who use modern contraception are 5.4 times more likely than their counterparts to use cervical cancer screening (AOR = 4.5; 95% CI: 1.8–16.3). Women who have a high level of knowledge regarding cervical cancer screening and its predisposing factors were 5.9 times more likely to utilize cervical cancer screening as compared to women who have a low level of knowledge regarding cervical cancer screening and its predisposing factors (AOR = 5.9; 95% CI: 2–17) (Table 3).

**Table 2. Reproductive health related factors of women in the age group of 30–49 years in Ambo town, Oromia Regional State, Ethiopia, February to March 2020.**

| Reproductive health related factors of study participants (n = 195) | Frequency | |
| --- | --- | --- |
| | Number/percentage of cases (n = 65) | Number/percentage of controls (n = 130) |
| **Age at first menses (years)** | | |
| <15 | 51 (78.5) | 98 (75.4) |
| ≥15 | 14 (21.5) | 32 (24.6) |
| **First coitrache** | | |
| <18 | 14 (21.5) | 19(15.7) |
| ≥ 18 | 51 (78.5) | 102 (84.3) |
| **Parity** | | |
| <5 | 40 (65.6) | 97 (89.8) |
| ≥5 | 21(34.4) | 11 (10.2) |
| **Age at first delivery** | | |
| 13–19 | 20 (32.8) | 21(19.4) |
| 20–24 | 38 (62.3) | 69 (63.9) |
| ≥25 | 3 (4.9) | 18 (16.7) |
| **Used modern contraceptives** | | |
| Yes | 51 (78.5) | 75 (57.7) |
| No | 14 (21.5) | 55 (42.3) |
| **Currently Your husband has another wife** | | |
| Yes | 3 (5.4) | 8 (7.2) |
| No | 53 (94.6) | 102 (92.7) |
| **Used condom** | | |
| Yes | 3 (4.6) | 5 (3.9) |
| No | 62 (95.4) | 124 (96.1) |
| **Number of partner in your life time** | | |
| 1 | 58 (89.2) | 110 (90.9) |
| ≥2 | 7 (10.8) | 11 (9.1) |

## Discussion

Routine cervical cancer screening is critical and the most effective method for early detection and treatment of precancerous lesions and mortality reduction from cervical cancer. This study was conducted to identify determinants of utilization of cervical cancer screening among women in Ambo Town, Central Ethiopia. Thus, this study identified factors like women's age, parity, use of modern contraception, and level of knowledge as determinants of utilization of cervical cancer screening.

The result of this study showed that women in the age group of 30–34 years were 0.2 times less likely to utilize cervical cancer screening as compared to those women in the age group of 40 years and above. This result was consistent with the studies conducted in Ethiopia, Dessie town, Debre Markos town, Finote Selam city, Zambia, Kenya, and rural areas of Mexico, which revealed that women in the younger age groups were less likely to utilize cervical cancer screening as compared with those women in the older age groups [18, 35–39]. A possible

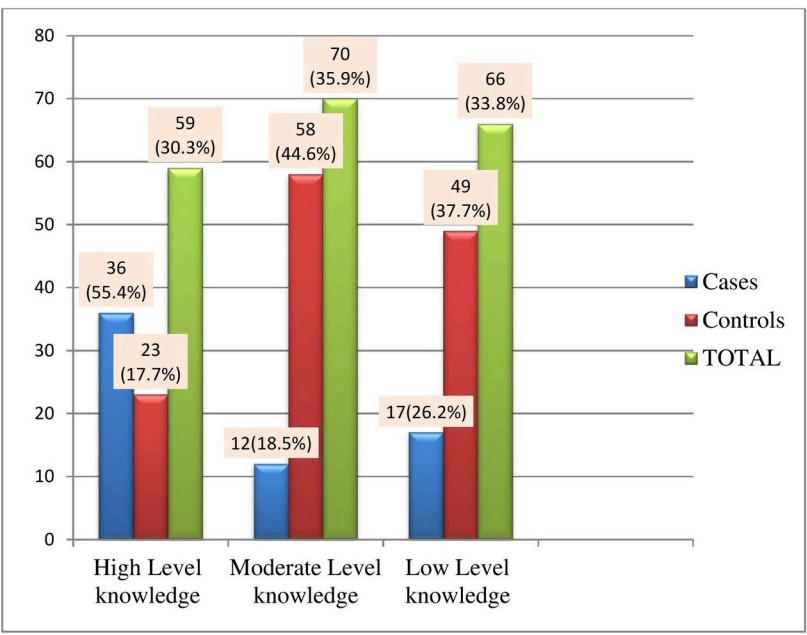

**Fig 1. Knowledge of cervical cancer screening among women in the age group of 30–49 years in Ambo town, Oromia Regional State, Ethiopia, February to March, 2020.**

reason why women in the younger age groups were less likely to utilize cervical cancer screening is that they might consider themselves low-risk groups, and cancer-related morbidity and mortality are diseases of older age groups, which might be screened in their 30's and above. Similarly, the explanation for this could be that of the bimodal distribution of cervical cancer, one in their 30s and the other in their 60s. These two age groups represent the ages at which cervical lesions become symptomatic.

Consequently, women see themselves as being at an increased risk of invasive cervical cancer as their age increases and seek health care and cervical cancer screening services. Additionally,

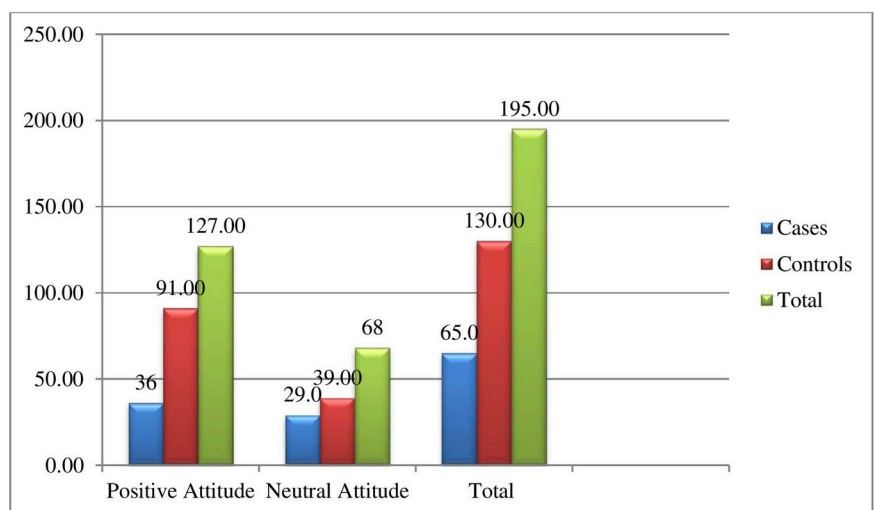

**Fig 2. Attitude of cervical cancer screening among women in the age group of 30–49 years in Ambo town, Oromia Regional State, Ethiopia, February to March, 2020.**

**Table 3. Determinant of utilization of cervical cancer screening among women in the age group of 30–49 in Ambo town, Oromia Regional State, Ethiopia.**

| Variables | Status of utilization of cervical cancer screening among women in the age group of 30–49 Years | | | | |
|---|---|---|---|---|---|
| | Cases | Controls | COR(95%)CI | AOR(95%CI) | P-Value |
| **Age** | | | | | |
| 30–34 | 19 | 80 | 0.24(0.1–0.5) | **0.2(0.06–0.7)***  | **0.010** |
| 35–39 | 29 | 33 | 0.88(0.38–2.03) | 0.9(0.3–2.8) | 0.817 |
| ≥40 | 17 | 17 | 1.00 | 1.00 | |
| **Educational status** | | | | | |
| Illiterate | 14 | 17 | 1.00 | 1.00 | |
| Primary (grade1-8) | 14 | 17 | 0.63(0.24–1.64) | 0.8(0.2–3.5) | 0.814 |
| Secondary(grade 9–12) | 13 | 27 | 0.59(0.22–1.54) | 0.9(0.2–4.4) | 0.879 |
| Diploma and above | 24 | 59 | 0.49(0.21–1.12) | 0.3(0.04–1.8) | 0.173 |
| **Occupation** | | | | | |
| Unemployed | 37 | 56 | 1.00 | 1.00 | |
| Governmental employee | 21 | 49 | 0.7(0.3–1.3) | 2.4(0.5–11.6) | 0.278 |
| Self-employee | 7 | 25 | 0.4(0.2–1.1) | 0.7(0.18–2.8) | 0.626 |
| **Monthly income** | | | | | |
| <1500 ETB | 22 | 30 | 1.00 | 1.00 | |
| 1501–3000 ETB | 7 | 20 | 0.48(0.17–1.33) | 0.5(0.1–2.4) | 0.422 |
| >3000 ETB | 36 | 80 | 0.6(0.31–1.21) | 1.2(0.4–4.0) | 0.731 |
| **Parity** | | | | | |
| <5 | 40 | 97 | 1.00 | 1.00 | |
| ≥5 | 21 | 11 | 4.6(2–10.5) | **4.5(1.4–14.1)***  | **0.009** |
| **Use of condom** | | | | | |
| Yes | 3 | 5 | 1.2(0.3–5.2) | | 0.807 |
| No | 62 | 124 | 1.00 | 1.00 | |
| **Use of modern contraception** | | | | | |
| Yes | 51 | 75 | 2.7(1.3–5.3) | **5.4(1.8–16.3)***  | **0.003** |
| No | 14 | 55 | 1.00 | 1.00 | |
| **Age at first delivery** | | | | | |
| 13–19 | 20 | 21 | 5.7(1.5–22.4) | 7.8(0.9–6.5) | 0.057 |
| 20–24 | 38 | 69 | 3.3(0.9–11.9) | 4.5(0.9–22.5) | 0.070 |
| ≥25 | 3 | 18 | 1.00 | 1.00 | |
| **Level of Knowledge** | | | | | |
| Low level | 17 | 49 | 1.00 | 1.00 | |
| Moderate level | 12 | 58 | 0.6(0.3–1.4) | 0.5(0.2–1.6) | 0.249 |
| High level | 36 | 23 | 4.5(2.1–9.7) | **5.93(2.0–17.0)***  | **0.001** |
| **Level of Attitude** | | | | | |
| Neutral attitude | 91 | 29 | 1.00 | 1.00 | |
| Positive attitude | 36 | 39 | 0.5(0.28–0.99) | 1.1(0.4–3.1) | 0.802 |

Case = women aged 30–49 screened for cervical cancer, Control = women aged 30–49 not screened for cervical cancer, Crude odds ratio (COR), Adjusted odds ratio (AOR), Confidence interval (CI),

*p < 0.05.

in Ethiopia, the cervical cancer screening guideline promotes women aged 30–49 to be screened for cervical cancer, and women aged 40 and above might have better health-seeking behaviour and intention to be screened. Furthermore, this age group is more susceptible to giving birth at a productive age and has a chance of getting more gynaecological examinations, giving birth, and getting more health information about sexual and reproductive health, including cervical cancer screening services. The other explanation might also be that increasing risk with women's age leads the women to have more contact with healthcare facilities.

However, the study conducted in Ethiopia's Tigray region public hospitals and India indicated that women in the younger age groups were more likely to utilize cervical cancer screening than women in the older age groups [28, 40]. The possible reasons for the discrepancy in the results might be due to the variation in the study participants, time deference', availability of information, and freedom of access to information regarding cervical cancer screening and its predisposing factors through social media and other routes.

The findings of this study revealed that being parity five or above among women was 4.5 times more likely to utilize cervical cancer screening when compared to being less than parity five. The result of this study was comparable with the study findings reported from Arba Minch town, southern Ethiopia, Tanzania, Dare Salaam, and India, which showed that women with a history of more parity were more likely to utilize cervical cancer screening [40–43]. This may be due to repeated visits to healthcare facilities for family planning, deliveries, and antenatal care follow-up so that they may get advice to use the service and also receive screening during their early deliveries.

In contrast, the study conducted at Finote Salam, Northwest Ethiopia and Jamaica revealed that women with a history of more parity were less likely to utilize cervical cancer screening as compared with women with a history of less parity [30, 43]. Differences in respondents' ages, levels of awareness, access to information such as mass media and other social media, family, peers, cultural beliefs, sociodemographic status, women's autonomy, economic conditions, physical and financial accessibility, disease patterns, and health service issues, as well as differences in the study design, study area, study period, study populations, and sample size, all contribute to the variation.

The result of this study showed that women who use modern contraception were 5.4 times more likely to utilize cervical cancer screening as compared with their counterparts. The findings of this study are consistent with the study conducted in Jimma Town, Southwest Ethiopia, Burkina Faso, Malawi, and India, which indicated that those women who used modern contraceptives were more likely to utilize cervical cancer screening as compared to their counterparts [40, 44–46]. This could be as a result of customers receiving counseling on cervical cancer screening and predisposing factors while receiving family planning services.

Women who had a high level of knowledge regarding cervical cancer screening and its predisposing factors were 5.9 times more likely to utilize cervical cancer screening as compared to women who had a low level of knowledge regarding cervical cancer screening and its predisposing factors. The finding of this study is consistent with the study conducted in Malawi, Mexico, Ghana, and Nairobi, Kenya, which revealed that women who have a high level of knowledge are more likely to utilize cervical cancer screening as compared to their counterparts [38, 39, 46, 47]. Furthermore, the findings of this study are similar to those of studies conducted in other parts of Ethiopia, including Arba Minch Town, Addis Ababa, and Jimma Town [25, 32, 41].

## Conclusion

The age of women, parity, use of modern contraception, and level of knowledge regarding cervical cancer screening and its predisposing factors were the determinants of the utilization of

cervical cancer screening among women. As a result, the media, the health bureau, and health professionals should advocate raising awareness about cervical cancer and its preventative methods, which are primarily focused on screening.

## Strength and limitation of the study

To account for contextual variance in the research participants, cases and controls were recruited from the same neighborhood.

Because of the long time after the event, recall bias exists for queries like "first menses" or "first coitrache."

Even though data collectors were trained to maintain as much privacy as possible to boost respondents' confidence and promote their responses, some respondents may withhold some information to give socially acceptable answers to particular questions.

## Supporting information

**S1 Data. Cervical cancer screening raw data.**
(SAV)

## Acknowledgments

We would like to thank the study participants and all other people who were formally or informally involved in the accomplishment of this research.

**Ethical approval and consent to participate**

An ethical clearance was obtained from the Ethical Review Board of Ambo University College of Medicine and Health Sciences, with the reference number PGC/051/2019. Hierarchically, all administrative bodies communicated and permission was secured. Written informed consent was obtained from the study subjects after explaining the objectives and procedures of the study and their right to participate or to withdraw at any time during the interview. The Research and Ethical Review Committee also approved its ethical issues as there was no procedure that affected the study subject and the data was used only for research purposes. For this purpose, a one-page consent letter was attached to the cover page of each questionnaire, stating the general purpose of the study and issues of confidentiality, which were discussed by data collectors before proceeding to the interview. Finally, we certify that this study was conducted in accordance with the Helsinki Declaration.

## Author Contributions

**Conceptualization:** Dereje Lemma, Mecha Aboma, Teka Girma, Abebe Dechesa.

**Data curation:** Dereje Lemma, Mecha Aboma, Teka Girma, Abebe Dechesa.

**Formal analysis:** Dereje Lemma, Mecha Aboma, Teka Girma, Abebe Dechesa.

**Funding acquisition:** Dereje Lemma, Mecha Aboma, Teka Girma, Abebe Dechesa.

**Investigation:** Dereje Lemma, Mecha Aboma, Teka Girma, Abebe Dechesa.

**Methodology:** Dereje Lemma, Mecha Aboma, Teka Girma, Abebe Dechesa.

**Project administration:** Dereje Lemma, Mecha Aboma, Teka Girma, Abebe Dechesa.

**Resources:** Dereje Lemma, Mecha Aboma, Teka Girma, Abebe Dechesa.

**Software:** Dereje Lemma, Mecha Aboma, Teka Girma, Abebe Dechesa.

**Supervision:** Dereje Lemma, Mecha Aboma, Teka Girma, Abebe Dechesa.

**Validation:** Dereje Lemma, Mecha Aboma, Teka Girma, Abebe Dechesa.

**Visualization:** Dereje Lemma, Mecha Aboma, Teka Girma, Abebe Dechesa.

**Writing – original draft:** Dereje Lemma, Mecha Aboma, Teka Girma, Abebe Dechesa.

**Writing – review & editing:** Dereje Lemma, Mecha Aboma, Teka Girma, Abebe Dechesa.

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
