## [Decision Letter · Decision Letter 0]

15 Mar 2022

PONE-D-21-40695Determinants of Utilization of Cervical Cancer Screening among Women in the Age group of 30-49 Years in Ambo Town, Central Ethiopia, a case-control studyPLOS ONE

Dear Mecha Aboma,

Thank you for submitting your manuscript to PLOS ONE. After careful consideration, we feel that it has merit but does not fully meet PLOS ONE’s publication criteria as it currently stands. Therefore, we invite you to submit a revised version of the manuscript that addresses the points raised during the review process.

We look forward to receiving your revised manuscript.

Kind regards,

Desalegn Admassu Ayana, Ph.D

Academic Editor

PLOS ONE

Journal Requirements:

“This research received no specific grant from any funding agency in the public, commercial or not-for-profit sectors.”

4. Please include a separate caption for each figure in your manuscript

Reviewers' comments:

Reviewer's Responses to Questions

**Comments to the Author**

1. Is the manuscript technically sound, and do the data support the conclusions?

Reviewer #1: Partly

Reviewer #2: Yes

2. Has the statistical analysis been performed appropriately and rigorously? 

Reviewer #1: Yes

Reviewer #2: Yes

3. Have the authors made all data underlying the findings in their manuscript fully available?

Reviewer #1: Yes

Reviewer #2: Yes

4. Is the manuscript presented in an intelligible fashion and written in standard English?

Reviewer #1: No

Reviewer #2: Yes

5. Review Comments to the Author

Reviewer #1: Many grammatical errors are observed in the manuscript, the study is a case control study but in the conclusion section it is reported "Even though the facility of cervical cancer screening is available in almost all health facility levels but only a few of them utilized the services from existing data" this seems reporting prevalence of cervical cancer screening.

Reviewer #2: The manuscript is reasonably sound, and the data support the conclusions presented. However, there are matters require revisions. The statistical analysis seems rigorous. The authors reported availability of data and promised that provide the data with reasonable request. The manuscript requires language revision.

General comments:

This article tried to address important research question that is relevant and an interesting. The manuscript is generally well written and structured. However, in my opinion the paper has few shortcomings. For instance, research gap that authors would attempt to address in their project is not boldly stated.

It needs to show gap and must attempt to fill in some piece of information missed in the scientific literature. Otherwise, it is not novel research which is not contributing to the overall goals of science.

Title: Determinants of Utilization of Cervical Cancer Screening among Women in the Age group of 30-49 Years in Ambo Town, Central Ethiopia: a case-control study.

Why would you (authors) focus on this age group (30-49 years of age) as far as 15-49 years are also eligible?

ABSTRACT:

Background: What stated in this section might not be research gap. It seems effort attempted by government to facilitate cervical cancer screening. What were known and not known since guidelines and protocols endorsed by government of Ethiopia?

Methods: “multivariate” should be edited and corrected as “Multivariable”

Results: it would be clearer if the total number of study participants involved were mentioned.

Line 20. Is grand multiparity appropriate instead of saying para 5 and above?? It is optional.

Line 22: high level of knowledge can be merged to the list of variables showed association with CCS. No need of separate explanation.

Conclusions: I think it needs revision. I must be based on main findings.

Line 27: Why social media? Or would the authors say mass media? How many of women have access to social media? How feasible?

INTRODUCTION:

Line 31: It would be better if it changed to “introduction” than background. Because introduction consists of background and statement of the problem.

Line 37-40: these statements are not aligning with the prior statements and subsequent paragraph. The first paragraph state prevalence and death attributed to cervical Ca. The middle statements talking about methods of screening while the subsequent paragraph talking about Cervical Ca incidence by regions.

Similarly, lines 103-108 are not appropriate place for this paragraph. Check it.

Line 111: Dire Dawa is not a city administration. Rather, it is simply Dire Dawa Administration since it has the numbers of rural woredas and kebeles.

METHODS and MATERIALS:

Study setting- was missed.

This subtitle is important for what the authors stated from line 124-129. In addition, I personally want to incorporate other pertinent information such the numbers of health facilities, and trained health care providers who can provide CCS.

Data collection tool and personnel

Line 154-158: Here detail expiations about number of items used, response options, how composite scores made and relevant nature of tools, their validity are required. I suggest that merging and well synthesizing this section with operational definitions.

Line 189: there is phrase “overall knowledge”. What does it mean? Did the knowledge items have dimensions or domains? If that is the case define overall and dimensions too.

Data management & Analysis: add versions of software used (EpiData and SPSS)

RESULTS:

Line 215-216; Either median with IQR or mean with SD is enough based distribution of the data.

Table 1: replace ‘’Socio-demographic characteristics of

study participants (n=195)’’ with “variables”.

Line 226 and 229: replace “have seen” with “saw” and “started” respectively.

Line 234-235 remove “were”

Line 237: replace “have” with “had”

Table 2: variable “your husband has another wife”. It is not clear whether it is talking about polygamy or another wife or partner before she married him??

Table 2: “Number of partners”. It is not clear. Was it asking about number of partners in life or current number of partners?

Knowledge related factors:

Line 242-245: The statements are too long, and vague/not clear. It was also mentioned under operational definitions that authors followed Bloom’s cut-off point to determine levels of knowledge. The same is true in lines 249-251.

Lines 255-265 need revision. Things already addressed in method section should be omitted. E.g., criteria set for candidacy to multivariable analysis.

Lines 270, 273, and 275: The phrases “in the age group 30-49” are presented redundantly here and there unnecessarily. Since it was already mentioned in the title, no need of reporting them. Simply use “women who…….”

Line 279: “Towns”- delete “s”. Similarly, delete February to march” since it was mentioned in line 125.

Table 3: under monthly income (>3000ETB) the corresponding confidence interval was 0.4-4. But to maintain consistency use at least one digit after decimal point. The same is true for high level of knowledge (2-17). It would be better it 2.0-17.0 if appropriate or subsequent number.

DISCUSSION:

It would be appropriate if the first paragraph of discussion present summary of main findings. The main purpose of discussion is reporting findings one by one, interpreting, comparing with former findings and suggesting possible scientific justifications. In addition, explain newly emerged understandings from the current study.

This section needs strict revision, and arguments.

Line 292. It indicates misperception among younger women. Why that? Authors’ argument/ justifications should be scientific and strong enough to convince. Again, there are inconclusive findings? It is suggested that this must be exhaustively addressed and recommendations might be also required.

Line 301-308: it is wonderful ways of discussion.

Line 304: Add Tanzania, after Dare Salam.

Lines 311-313: These kind of stating justifications might be so superficial. It would better if authors specify reasons for difference than crudely reporting difference was due to “socio-demographic or sample size, time gaps”. Please, look them further in terms of design, inclusion criteria, age, awareness level etc.

CONCLUSIONS:

The first 3 lines (331-333) are enough. Recommendations are not appropriately made based main findings.

For instance, statements in line 334 and 335(availability of the service was not studied in the current study. Three paragraphs starting from line 336-348 have no importance.

FIGURES:

Figure 1: has no importance. I would suggest to omit it.

Figure 3: has miss classification in relation to what operationally defined.

REFERENCES:

• There is inconsistent way of reference writing. Some lacks volume, numbers, years etc.

• There are also old references published 10 years back. Unless special circumstances, it would better to use articles published since 2011 if guideline of journal allow.

OTHERS:

Minor editorial: punctuations, grammatical errors, inconsistent use terms

NOTE: in conclusion, the findings of the manuscript are informative. But it requires moderate revision.

6. PLOS authors have the option to publish the peer review history of their article (what does this mean?). If published, this will include your full peer review and any attached files.

Reviewer #1: No

Reviewer #2: No

---

## [Author Response · Author response to Decision Letter 0]

8 Apr 2022

Author's response to Reviewers Comments

Manuscript ID number: PONE-D-21-40695 

Title of Manuscript: Determinants of Utilization of Cervical Cancer Screening among Women in the Age group of 30-49 Years in Ambo Town, Central Ethiopia, a case-control study

Authors: 

1. Dereje Lema

2. Mecha Aboma: Corresponding Author: abomamecha@gmail.com

3. Teka Girma 

4. Ababe Dechasa

Generally, in manuscript we used track change to incorporate or address the reviewers’ comments.

A Point by Point Response to Reviewer Evaluation 

1. Reviewer Question, #1: Many grammatical errors are observed in the manuscript, the study is a case control study but in the conclusion section it is reported "Even though the facility of cervical cancer screening is available in almost all health facility levels but only a few of them utilized the services from existing data" this seems reporting prevalence of cervical cancer screening.

Answers from authors: The comment is accepted and corrected using track change and colored highlight throughout the manuscript, Line 407-4025

2. Reviewer Question, #2: The manuscript is reasonably sound, and the data support the conclusions presented. However, there are matters require revisions. The statistical analysis seems rigorous. The authors reported availability of data and promised that provide the data with reasonable request. The manuscript requires language revision.

Answers from authors: It is alright. As you indicate there are many errors. Using language expert we tried to correct using track change and colored highlight throughout the manuscript 

3. Reviewer Question, General comments: This article tried to address important research question that is relevant and an interesting. The manuscript is generally well written and structured. However, in my opinion the paper has few shortcomings. For instance, research gap that authors would attempt to address in their project is not boldly stated.

It needs to show gap and must attempt to fill in some piece of information missed in the scientific literature. Otherwise, it is not novel research which is not contributing to the overall goals of science. 

Answers from authors: The comment is accepted and correction and revisions is made using track change and colored highlight throughout the manuscript.

4. Reviewer Question, Title: Determinants of Utilization of Cervical Cancer Screening among Women in the Age group of 30-49 Years in Ambo Town, Central Ethiopia: a case-control study. Why would you (authors) focus on this age group (30-49 years of age) as far as 15-49 years are also eligible?

Answers from authors: The government of Ethiopia launched screening of cervical cancer in 2014 and the VIA recommended for those women between the age of 30-49 years within every five years (6). Currently in Ethiopia cervical cancer screening services are free of charge in the public health facilities and women of 30-49 can uses services through self-referral, referral by health extension workers, by referral of other health care professionals (nurses, midwives, doctors ,and public health professionals) , and opportunistic screening.

5. Reviewer Question, ABSTRACT: Background: What stated in this section might not be research gap. It seems effort attempted by government to facilitate cervical cancer screening. What were known and not known since guidelines and protocols endorsed by government of Ethiopia?

Answers from authors: The comment is accepted and corrected, and indicated by track change and colored highlight, Line 11-16

6. Reviewer Question, Methods: “multivariate” should be edited and corrected as “Multivariable” 

Answers from authors: The comment is accepted and corrected as “Multivariable”, Line 20

7. Reviewer Question, Results: it would be clearer if the total number of study participants involved were mentioned.

Answers from authors: The comment is accepted and corrected as ‘‘a total of 195 study participants, sixty-five cases and one hundred thirty controls, were participated in this study making a response rate of 100%’’, Line 22-23

8. Reviewer Question, Line 20. Is grand multiparty appropriate instead of saying Para 5 and above?? It is optional.

Answers from authors: The comment is accepted 

9. Reviewer Question, Line 22: high level of knowledge can be merged to the list of variables showed association with CCS. No need of separate explanation. 

Answers from authors: The comment is accepted and corrected and indicated by using track change and colored highlight Line 25-26

10. Reviewer Question, Conclusions: I think it needs revision. It must be based on main findings.

Answers from authors: The comment is accepted and revision is made and indicated by using track change and colored highlight, Line 407-4025

11. Reviewer Question, Line 27: Why social media? Or would the authors say mass media? How many of women have access to social media? How feasible?

Answers from authors: The comment is accepted and corrected as ‘‘Mass media’’ Line 31

INTRODUCTION: 

12. Reviewer Question, Line 31: It would be better if it changed to “introduction” than background. Because introduction consists of background and statement of the problem.

Answers from authors: The comment is accepted and corrected as ‘‘Introduction’’, Line 35

13. Reviewer Question, Line 37-40: these statements are not aligning with the prior statements and subsequent paragraph. The first paragraph state prevalence and death attributed to cervical Ca. The middle statements talking about methods of screening while the subsequent paragraph talking about Cervical Ca incidence by regions. Similarly, lines 103-108 are not appropriate place for this paragraph. Check it.

Answers from authors: The comment is accepted and revision is made and indicated by using track change colored highlight throughout the manuscript

14. Reviewer Question, Line 111: Dire Dawa is not a city administration. Rather, it is simply Dire Dawa Administration since it has the numbers of rural woredas and kebeles.

Answers from authors: The comment is accepted and corrected as ‘‘Dire Dawa Administration’’, Line 147

15. Reviewer Question, METHODS and MATERIALS:

Study setting- was missed.

This subtitle is important for what the authors stated from line 124-129. In addition, I personally want to incorporate other pertinent information such the numbers of health facilities, and trained health care providers who can provide CCS.

Answers from authors: The comment is accepted and corrected as: - ‘‘The town has one referral hospital, one general hospital, two health centers, nine health posts, and twenty one private clinics. Ambo General Hospital was the only hospital providing cervical cancer screening during this study was conducted. Ambo General Hospital had 287 health care workers. Those were 8 special doctors, 19 general practitioners, 69 nurses, and other health care providers (30).’’ Line 168-172. And study setting is added. Line 62

Data collection tool and personnel

16. Reviewer Question, Line 154-158: Here detail expiations about number of items used, response options, how composite scores made and relevant nature of tools, their validity are required. I suggest that merging and well synthesizing this section with operational definitions.

Answers from authors: The comment is accepted and revision is made and indicated by using track change and colored highlight. Number of items also addressed under operational definition., Line 198-204

17. Reviewer Question, Line 189: there is phrase “overall knowledge”. What does it mean? Did the knowledge items have dimensions or domains? If that is the case define overall and Dimensions too.

Answers from authors: The comment is accepted and revision is made and indicated by using track change and colored highlighted, Line 235-239

18. Reviewer Question, Data management & Analysis: add versions of software used (EpiData and SPSS)

Answers from authors: The comment is accepted and corrected as ‘‘EPI-Data 3.1 version SPSS software version 25’’, Line 2013-2014

RESULTS:

19. Reviewer Question, Line 215-216; Either median with IQR or mean with SD is enough based distribution of the data 

Answers from authors: The comment is accepted

20. Reviewer Question, Table 1: replace ‘’Socio-demographic characteristics of study participants (n=195)’’ with “variables”.

Answers from authors: The comment is accepted and indicated by using track change and colored highlighted, and characteristics replaced with ‘‘variables’’

21. Reviewer Question, Line 226 and 229: replace “have seen” with “saw” and “started” respectively.

Answers from authors: The comment is accepted and indicated by using track change and colored highlighted, and “have seen” replaced with “saw” and “started” respectively.’’ Line 273, 276

22. Reviewer Question, Line 234-235 remove “were

Answers from authors: The comment is accepted and corrected, and indicated by using track change and colored highlighted ‘’were’’ is removed, Line 281, 282

23. Reviewer Question, Line 237: replace “have” with “had”

Answers from authors: The comment is accepted corrected, and indicated by using track change and colored highlighted, and “have” replaced with “had” , Line 284

24. Reviewer Question, Table 2: variable “your husband has another wife”. It is not clear whether it is talking about polygamy or another wife or partner before she married him??

Answers from authors: the comments accepted and corrected, and indicated by using track change and colored highlighted. It is talking about polygamy or currently having another wife 

25. Reviewer Question, Table 2: “Number of partners”. It is not clear. Was it asking about number of partners in life time or current number of partners?

Answers from authors: The comment is accepted and corrected, and indicated by using track change and colored highlighted. It is talking about number of partners in life time 

26. Reviewer Question, Knowledge related factors: Line 242-245: The statements are too long, and vague/not clear. It was also mentioned under operational definitions that authors followed Bloom’s cut-off point to determine levels of knowledge. The same is true in lines 249-251.

Answers from authors: The comment is accepted and correction is made and indicated by using track change and colored highlighted, Line 290-292, 296-298

27. Reviewer Question, Lines 255-265 need revision. Things already addressed in method section should be omitted. E.g., criteria set for candidacy to multivariable analysis.

Answers from authors: The comment is accepted and correction is made and indicated by using track change and colored highlight, Line 302-3013

28. Reviewer Question, Lines 270, 273, and 275: The phrases “in the age group 30-49” are presented redundantly here and there unnecessarily. Since it was already mentioned in the title, no need of reporting them. Simply use “women who…….” 

Answers from authors: The comment is accepted and correction is made and indicated by using track change and colored highlight, Line 322, 324

29. Reviewer Question, Line 279: “Towns”- delete “s”. Similarly, delete February to march” since it was mentioned in line 125.

Answers from authors: The comment is accepted and correction is made and indicated by using track change and colored highlight, Line 328

30. Reviewer Question, Table 3: under monthly income (>3000ETB) the corresponding confidence interval was 0.4-4. But to maintain consistency use at least one digit after decimal point. The same is true for high level of knowledge (2-17). It would be better it 2.0-17.0 if appropriate or subsequent number.

Answers from authors: The comment is accepted and correction is made and indicated by using track change and colored highlight

31. Reviewer Question, DISCUSSION: It would be appropriate if the first paragraph of discussion present summary of main findings. The main purpose of discussion is reporting findings one by one, interpreting, comparing with former findings and suggesting possible scientific justifications. In addition, explain newly emerged understandings from the current study.

Answers from authors: The comment is accepted and revision is made and indicated by using track change and colored highlight throughout the manuscript, Line 334-338

32. Reviewer Question, This section needs strict revision, and arguments. Line 292. It indicates misperception among younger women. Why that? Authors’ argument/ justifications should be scientific and strong enough to convince. Again, there are inconclusive findings? It is suggested that this must be exhaustively addressed and recommendations might be also required.

Answers from authors: The comment is accepted and revision is made and indicated by using track change and colored highlight as follow: - ‘‘Similarly, the explanation for this could be that of the bimodal distribution of cervical cancer, one in their 30s and the other in their 60s. These two age groups represent the ages at which cervical lesions become symptomatic. 

Consequently, women see themselves as being at an increased risk of invasive cervical cancer as their age increases and seek health care and cervical cancer screening services. Additionally, in Ethiopia, the cervical cancer screening guideline promotes women aged 30–49 to be screened for cervical cancer, and women aged 40 and above might have better health-seeking behaviour and intention to be screened. Furthermore, this age group is more susceptible to giving birth at a productive age and has a chance of getting more gynecological examinations, giving birth, and getting more health information about sexual and reproductive health, including cervical cancer screening services. The other explanation might also be that increasing risk with women’s age leads the women to have more contact with healthcare facilities.’’ Line 346-459

33. Reviewer Question, Line 301-308: it is wonderful ways of discussion.

Answers from authors: I thank you!

34. Reviewer Question, Line 304: Add Tanzania, after Dare Salam.

Answers from authors: The comment is accepted and Tanzania is added after Dare Salam Line 369

35. Reviewer Question, Lines 311-313: These kinds of stating justifications might be so superficial. It would better if authors specify reasons for difference than crudely reporting difference was due to “socio-demographic or sample size, time gaps”. Please, look them further in terms of design, inclusion criteria, age, awareness level etc.

Answers from authors: The comments accepted and revision is made and indicated by using track change and colored highlight as follow: Differences in respondents' ages, levels of awareness, access to information such as mass media and other social media, family, peers, cultural beliefs, sociodemographic status, women's autonomy, economic conditions, physical and financial accessibility, disease patterns, and health service issues, as well as differences in the study design, study area, study period, study populations, and sample size, all contribute to the variation. Line 377-383

36. Reviewer Question, CONCLUSIONS: The first 3 lines (331-333) are enough. Recommendations are not appropriately made based main findings. For instance, statements in line 334 and 335(availability of the service was not studied in the current study. Three paragraphs starting from line 336-348 have no importance.

Answers from authors: The comment is accepted and correction is made and indicated by using track change and colored highlight, Line 405-425

FIGURES:

37. Reviewer Question, Figure 1: has no importance. I would suggest to omit it.

Answers from authors: The comment is accepted and the figure is removed

38. Reviewer Question, Figure 3: has miss classification in relation to what operationally defined.

Answers from authors: The comment is accepted but the score/ or response for negative attitude was zero because all of the study participants has positive and neutral attitudes, (according to their response indicated) 

39. Reviewer Question, REFERENCES: 

• There is inconsistent way of reference writing. Some lacks volume, numbers, years etc.

• There are also old references published 10 years back. Unless special circumstances, it would better to use articles published since 2011 if guideline of journal allow.

Answers from authors: The comment is accepted, correction and revision is made and indicated by using track change and colored highlight

40. Reviewer Question, OTHERS: Minor editorial: punctuations, grammatical errors, inconsistent use terms

Answers from authors: The comment is accepted, correction and revision is made and indicated by using track change and colored highlight throughout the manuscript

41. Reviewer Question, NOTE: in conclusion, the findings of the manuscript are informative. But it requires moderate revision.

Answers from authors: The comment is accepted and revision is made and indicated by using track change and colored highlight throughout the manuscript.

---

## [Decision Letter · Decision Letter 1]

13 May 2022

PONE-D-21-40695R1Determinants of Utilization of Cervical Cancer Screening among Women in the Age Group of 30-49 Years in Ambo Town, Central Ethiopia: A Case-Control StudyPLOS ONE

Dear Dr. Aboma,

Thank you for submitting your manuscript to PLOS ONE. After careful consideration, we feel that it has merit but does not fully meet PLOS ONE’s publication criteria as it currently stands. Therefore, we invite you to submit a revised version of the manuscript that addresses the points raised during the review process.

We look forward to receiving your revised manuscript.

Kind regards,

Desalegn Admassu Ayana, Ph.D

Academic Editor

PLOS ONE

Journal Requirements:

Reviewers' comments:

Reviewer's Responses to Questions

**Comments to the Author**

1. If the authors have adequately addressed your comments raised in a previous round of review and you feel that this manuscript is now acceptable for publication, you may indicate that here to bypass the “Comments to the Author” section, enter your conflict of interest statement in the “Confidential to Editor” section, and submit your "Accept" recommendation.

Reviewer #1: All comments have been addressed

2. Is the manuscript technically sound, and do the data support the conclusions?

Reviewer #1: Yes

3. Has the statistical analysis been performed appropriately and rigorously? 

Reviewer #1: Yes

4. Have the authors made all data underlying the findings in their manuscript fully available?

Reviewer #1: Yes

5. Is the manuscript presented in an intelligible fashion and written in standard English?

Reviewer #1: Yes

6. Review Comments to the Author

Reviewer #1: All comments have been addressed by the author including the statistical analysis and the conclusions drawn from the data. Appropriate controls were chosen. As the author declares all the data were fully presented in the manuscript.

7. PLOS authors have the option to publish the peer review history of their article (what does this mean?). If published, this will include your full peer review and any attached files.

Reviewer #1: No

---

## [Author Response · Author response to Decision Letter 1]

20 May 2022

Author's response to Reviewers and Academic Editors Comments

Manuscript ID number: PONE-D-21-40695 

Title of Manuscript: Determinants of Utilization of Cervical Cancer Screening among Women in the Age group of 30-49 Years in Ambo Town, Central Ethiopia, a case-control study

Authors: 

1. Dereje Lema

2. Mecha Aboma: Corresponding Author: abomamecha@gmail.com

3. Teka Girma 

4. Ababe Dechasa

Generally, in manuscript we used track change to incorporate or address the reviewers’ comments.

A Point by Point Response to academic editor and reviewer(s) Evaluation 

1. Academic Editors Question, #1: Please review your reference list to ensure that it is complete and correct. If you have cited papers that have been retracted, please include the rationale for doing so in the manuscript text, or remove these references and replace them with relevant current references. Any changes to the reference list should be mentioned in the rebuttal letter that accompanies your revised manuscript. If you need to cite a retracted article, indicate the article’s retracted status in the References list and also include a citation and full reference for the retraction notice.

Answers from authors: The comment is accepted and corrected using track change and colored highlights throughout the manuscript. However, we didn’t use retracted and unpublished papers in the references, all of the references we used in this manuscript are available on Search Google or/and Google Scholars.

---

## [Editor Report · Decision Letter 2]

21 Jun 2022

Determinants of Utilization of Cervical Cancer Screening among Women in the Age Group of 30-49 Years in Ambo Town, Central Ethiopia: A Case-Control Study

PONE-D-21-40695R2

Dear Dr. Aboma,

We’re pleased to inform you that your manuscript has been judged scientifically suitable for publication and will be formally accepted for publication once it meets all outstanding technical requirements.

Kind regards,

Desalegn Admassu Ayana, Ph.D

Academic Editor

PLOS ONE
---

## [Editor Report · Acceptance letter]

30 Jun 2022

PONE-D-21-40695R2 

Determinants of Utilization of Cervical Cancer Screening among Women in the Age Group of 30-49 Years in Ambo Town, Central Ethiopia: A Case-Control Study 

Dear Dr. Aboma:

I'm pleased to inform you that your manuscript has been deemed suitable for publication in PLOS ONE. Congratulations! Your manuscript is now with our production department. 

Kind regards, 

on behalf of

Dr. Desalegn Admassu Ayana 

Academic Editor

PLOS ONE